# New ISE-Based Apparatus for Na^+^, K^+^, Cl^−^, pH and Transepithelial Potential Difference Real-Time Simultaneous Measurements of Ion Transport across Epithelial Cells Monolayer–Advantages and Pitfalls

**DOI:** 10.3390/s19081881

**Published:** 2019-04-20

**Authors:** Mirosław Zając, Andrzej Lewenstam, Magdalena Stobiecka, Krzysztof Dołowy

**Affiliations:** 1Department of Biophysics, Warsaw University of Life Sciences-SGGW, 159 Nowoursynowska St., 02-776 Warsaw, Poland; miroslaw_zajac@sggw.pl (M.Z.); magdalena_stobiecka@sggw.pl (M.S.); 2Centre for Process Analytical Chemistry and Sensor Technology (ProSens), Johan Gadolin Process Chemistry Centre, Åbo Akademi University, Biskopsgatan 8, 20500 Åbo-Turku, Finland; alewenst@abo.fi; 3Faculty of Materials Science and Ceramics, AGH University of Science and Technology, Mickiewicza 30, 30-059 Krakow, Poland

**Keywords:** ion-selective electrodes, epithelium, ion transport, cystic fibrosis

## Abstract

Cystic Fibrosis (CF) is the most common fatal human genetic disease, which is caused by a defect in an anion channel protein (CFTR) that affects ion and water transport across the epithelium. We devised an apparatus to enable the measurement of concentration changes of sodium, potassium, chloride, pH, and transepithelial potential difference by means of ion-selective electrodes that were placed on both sides of a 16HBE14σ human bronchial epithelial cell line that was grown on a porous support. Using flat miniaturized ISE electrodes allows for reducing the medium volume adjacent to cells to approximately 20 μL and detecting changes in ion concentrations that are caused by transport through the cell layer. In contrast to classic electrochemical measurements, in our experiments neither the calibration of electrodes nor the interpretation of results is simple. The calibration solutions might affect cell physiology, the medium composition might change the direction of actions of the membrane channels and transporters, and water flow that might trigger or cut off the transport pathways accompanies the transport of ions. We found that there is an electroneutral transport of sodium chloride in both directions of the cell monolayer in the isosmotic transepithelial concentration gradient of sodium or chloride ions. The ions and water are transported as an isosmotic solution of 145 mM of NaCl.

## 1. Introduction

Epithelial tissue that is made of a single cell layer bound together by tight junctions lines all the wet surfaces of the body. The tissue forms a barrier for the diffusive transport of water and solutes. Epithelial cells are polarized, i.e., they have different ion-transporting molecules on apical and basolateral faces. The difference in membrane compositions allows for the directional transport of ions. Water follows transported ions due to osmosis. The functioning of the ion transporters of epithelial cells is under external and intracellular control. The defect in ion and water transport is the cause of many different human disorders, including the most common fatal inherited one—cystic fibrosis (CF) [1]. It has been found that a defect in a single gene coding Cystic Fibrosis Transport Regulator (CFTR)—an anion channel that is present in the apical face of epithelial cell is responsible for CF [2,3]. While CF affects many organs in the human body, the most grievous effect is in the lungs, where a dense secretion that cannot be cleared from the airways leads to opportunistic infections and, in the pancreas, where dense mucus blocks the ducts, which results in lack of digestive enzymes and, in consequence, malnutrition. Around 70,000 people around the world suffer from CF.

The CFTR channel is not a major ion transporter of the cell membrane. There are many other channels, exchangers, and pumps (reviewed e.g., by [1,4,5,6,7,8]). Intracellular secondary messengers c-AMP and cell calcium concentration changes [9], activation of P2Y surface receptors [10], changes in cytoplasmic and extracellular pH [11,12], and cell volume change control the concerted actions of these transporting molecules.

There are channels of low conductivity in the apical face of the bronchial epithelium: the CFTR channel [13], the CaCC calcium-sensitive chloride channel [14], and the ENaC amiloride-sensitive sodium channel [15]. On the basolateral face, there are: the NaK2Cl co-transporter, which transports at the same time sodium, potassium and two chloride ions and the Na2HCO3 co-transporter, which transports in the same direction one sodium and two bicarbonate ions, as well as the NaKATPase pump, which transports three sodium cations out from the cell in exchange for two potassium cations into the cytoplasm. On both sides of the bronchial epithelium, there are: the NHE sodium for proton exchanger, the AE anion exchanger that transports the chloride and bicarbonate anions in opposite directions, the potassium channels [16], and the volume-sensitive anion channels [17].

Despite numerous scientists’ work on cystic fibrosis for many years, its mechanisms remain unknown. There are a few hypotheses concerning the defective water and ion transport in CF (too high NaCl concentration, too high water absorption, too low water secretion, insufficient bicarbonate secretion leading to mucin condensation, and apical surface liquid (ASL) acidification, etc.).

The study of ion transport across the epithelium was recently reviewed [18]. First, measurements of the ion transport across the frog skin go back to Ussing and Zerahn [19], who used radioactive ions. The radioisotope method allows for measurements of a single ion at 10-min intervals. The other possibility is the use of a voltage clamp or the current clamp technique across the tissue or cell monolayer in an Ussing chamber. The method has an obvious limitation. because the cations that are flowing inwards cannot be distinguished from the anions flowing outwards. Moreover, only the net current can be measured by means of this method.

Despite half a century of research, our knowledge of the ion and water transport across the epithelial cell layer is limited due to too many ion transporters, too many ions being transported i.e., chloride, bicarbonate, sodium, potassium, and hydrogen, and lack of a method that would allow for simultaneous measurements of all ions transported within very short time intervals (second time range).

Using ISE might be a reasonable choice to study ion transport across the epithelial cells. However, the use of ISE poses a serious technical problem. From the radiochemical measurements, one can estimate the sodium and chloride fluxes to be in the order of 1 μM∙cm^−2^∙h^−1^ = 280 pM∙cm^−2^∙s^−1^, and thus it will produce a change of ion concentration of only 28 nM∙s^−1^ in 10 ml medium, a change that is too small to be detected by ISE, except for the pH electrode [18]. Hug et al. [20] estimated bicarbonate secretion by epithelia using a pH electrode assuming that the change in pH is only caused by bicarbonate secretion, which is incorrect. Nair et al. [21] introduced silver wire that was coated with silver chloride 50 µm apart from the apical surface of epithelial cells that are known to secrete chloride anions. It was found that chloride is not the only ion that is secreted by the cells. Additionally, a calcium ISE electrode has been used to monitor calcium release from secreted mucin [22]. Our approach was different. We reduced the chamber volume to approximately 20–30 µl on both the apical and basolateral side of the epithelial cell layer. Thus, we could expect the change in ion concentration of 33–50 mM∙cm^−2^∙h^−1^, being readily measurable by ISE. We succeeded in constructing a device that measures the transport of four ions (sodium, potassium, chloride, and pH) across the epithelial cell monolayer grown on a porous support after a decade of tests and experiments [23,24]. The study provides insight into the mechanism of ion transport across the epithelium. 

## 2. Materials and Methods

### 2.1. The Apparatus

Figure 1 presents the measuring system. It consisted of a dismountable measuring platform that housed the electrodes, inlet and outlet tubes, and cells that were grown on a porous support of Snapwell insert (Corning Costar). Two syringe pumps (SP260PZ, WPI) that were connected to the measuring platform by silicone tubing exchanged the media. The silicone tubing that was located at the same heights to avoid hydrostatic pressure formation drained the excess fluid.

Two parts of the measuring platform body were cut from a polycarbonate sheet (PC) using a CNC machine tool and it was fitted to the size and shape of the Snapwell insert. Two O-rings were used to seal the insert on to the platform body. The distance between the housing and the porous membrane of the insert was less than 30 μm. The measuring platform was 40 mm wide, 40 mm long, and 32 mm thick. Seven holes of 3 mm diameter were directly drilled into the platform body against the cell layer. Soft silicone tubing was inserted into these holes and then glued to the platform body (3 mm OD and 1.5 mm ID for the ion-selective electrodes and 2 mm ID for the reference electrode) to form a washer. The stainless-steel inlet/outlet tubes were directly glued to the platform body. Electrodes were pushed into the washer before the experiments. The design allowed for the easy exchange of faulty electrodes.

The mounted measuring platform was placed in a Faraday cage to prevent external electromagnetic interferences. The ion-selective and reference electrodes were connected to a 16-channel Lawson Lab EMF Interface (Malvern, PA) that was connected to a PC computer with appropriate data acquisition software. 

### 2.2. The Cells

In the experiment, we used the human bronchial epithelial cell line (16HBE14 σ) that was kindly provided by the late Dr. Dieter Gruenert (UCSF). The cells were grown in Minimum Essential Medium supplemented with 10% fetal bovine serum, 200 mM l-Glutamine, and 10 μg/mL penicillin/streptomycin (Sigma) in a plastic Petri dish (Costar, Cambridge) coated with fibronectin/collagen/BSA incubated at 37 °C in the presence of 5% CO_2_. After 4–5 days, the cells reached 70–90% confluence and were detached by trypsinization (0.25% trypsin and 0.02% EDTA) and they were seeded onto Corning Costar Snapwell inserts (0.45 μm pore size, 1 cm^2^ surface area) at a density of 2.5·10^5^ cells/cm^3^. The cells were grown in submerged culture and the transepithelial resistance (TER) was checked every second day. The medium above the insert was aspired to form the air contact at the apical side of the monolayer to polarize the cells when the cells’ resistance reached 1000 Ω·cm^2^. The TER value started to decrease after establishing air contact. After 10–16 days, the cells were used in the experiment when the TER value reached 400 Ω·cm^2^. The cells that were used in the experiments were from the 5th to 25th passage after thawing. Figure 2 presents the schematic presentation of the cell culture procedure and the transepithelial resistance of the cell monolayers as a function of time.

### 2.3. The Media

We used isosmotic media that differed in sodium or chloride concentration to answer the question of whether the osmotic pressure difference across the cell monolayer was the only reason for ion and water transport. To maintain ionic strength and ionic activity in modified media, a choline cation replaced the sodium ion, and the chloride by a gluconate anion, respectively. Neither choline nor gluconate ions are transported across the epithelial layer. We used the Krebs–Henseleit solution (KHS1) and its two modifications. Table 1 provides their compositions. The KHS1 contains 141.8 mM of sodium and 129.1 mM of chloride ions. KHS2 24.8 mM of sodium ion and KHS3 contain 10.0 mM of chloride. All three solutions had the same ionic strength, ionic activity coefficient, and osmotic pressure (Π = 289 mOsm).

In some of the experiments, the KHS solutions were supplemented with 100 μM of amiloride—the potent apical Epithelial Sodium Channel (ENaC) blocker [25] and the blocker of sodium for proton exchanger (NHE), or 100 μM of 4,4’-Diisothiocyano-2,2’-stilbenedisulfonic acid (DIDS)—the blocker of anion channels and exchangers [26].

### 2.4. Electrodes

The 20 mm long micro-electrode body was made of the patch clamp borosilicate glass capillary (Harvard Apparatus). The compositions of membranes of ion-selective electrodes (ISEs) have been described earlier [24]. Briefly, the ISE membranes consisted of PVC (polyvinyl chloride), appropriate plasticizer and the ionophore was dissolved in freshly distilled THF (tetrahydrofuran). 2 µL of appropriate membrane solution was introduced to the 0.86 mm bore of the glass capillary to obtain the membrane, and the THF solvent was allowed to evaporate over 24 h. After the solvent had evaporated, the electrode bodies were filled with internal electrolyte solutions: 1% agar, 5% glycerine in 0.15 M KCl (for the potassium and chloride electrodes), or 0.15 M NaCl (for the sodium electrode), or 0.15 M KH_2_PO_4_-Na_2_HPO_4_ buffer pH 7.4 (for the pH electrode). Subsequently, the silver/silver chloride wire has sealed in the electrode. All of the chemicals were purchased from Sigma. We used a commercial reference microelectrode (ET069 Leakless Ag/AgCl Reference Electrode, eDAQ). 

### 2.5. Calibration of Electrodes

Each of the electrodes was separately calibrated before mounting on the measuring platform. Changing the appropriate ion concentration measured the calibration curves. The electrodes showed a linear relationship between the potential and log of ion activity. The slope for the chloride-sensitive electrode was s = –58 ± 2 mV/dec., for the potassium-sensitive electrode +55 ± 1 mV/dec., for the sodium +54 ± 1 mV/dec., and for the hydrogen +57 ± 2 mV/dec; Figure 3A shows the exemplary results for the chloride-sensitive electrode. The potential of the ion-selective electrodes was measured against a commercial macro-reference Ag/AgCl electrode (InLab, Mettler Toledo). The micro-reference electrodes that were used in our system differed from the commercial one by less than ±1 mV. The potential reproducibility and empirical calibration curve was determined in our system lacking a cell layer by repeating the medium exchange for different concentrations of the measured ion. Figure 3B presents the exemplary testing of the chloride-sensitive electrode in three KHS1 media lacking 10 mM of NaCl, the original KHS1, and KHS1 supplemented with 10 mM of NaCl. The same procedure was repeated for KHS2 and KHS3 media that were supplemented or lacking NaCl. All of the electrodes were tested by the same procedure. The media compositions for the sodium-sensitive electrode were the same as for the chloride-sensitive electrode, for the potassium-sensitive electrode, the media were supplemented with 2 mM of KCl (or lacking 2 mM of KCl), and for the pH-sensitive electrode, pH = 7.2, 7.4, and 7.6 were used.

The use of choline and gluconate ions have adverse effects on sodium-sensitive or chloride-sensitive ISEs. The slope for the chloride ISE in the presence of choline was decreased to s = −40 ± 2 mV/dec. in KHS2 medium and for the sodium ISE in the presence of gluconate to s = +40 ± 2 mV/dec. in KHS3 solution. The apparent sensitivity that was observed in KHS2 and KHS was encountered in the calculations of concentration changes in these solutions. 

ISEs are often used for the automated determination of ion concentrations in whole blood or plasma. In such measurements, the electrode is usually washed and calibrated before each measurement within a few seconds [27,28]. This could not be done in our experiment, since the washing solution is in direct contact with the cell layer, which influences its ion transport properties and the experiments last several minutes without the possibility of the medium change. Having access to different sodium ionophores: ETH 2120, ETH 4120, and Ionophore X, we decided to apply the first one. The selectivity for potassium of the Ionophore X is half an order better (in −log scale) than for the remaining ligands. However, we selected ETH 2120, since the membranes with this ionophore provide sufficient selectivity in the course of our experiments (−log K_Na,K_ = 1.5) and excellent stability over several months. The Na-ISE membrane with Ionophore X is not stable enough. It loses its properties after a few hours. Additionally, wide industrial application of the ETH 2120 ionophore in the environment of commercial analyzers supported our approach [28].

Therefore, the base-line potential stability check, reassembling the procedure known as one-point calibration, was employed. First, each ISE potential in the KHS1 solution was measured. Subsequently, after changing KHS1 to the other medium, the potential measured and the electrode slope were used to calculate the concentration (or concentration change) of a particular ion. The values of potential after 10 min were taken for the calculations. The base-line check was closed repeating the measurement in KHS1, see Figure 3C for a graphical illustration. The KHS1 solution is similar to the one in which the cells were grown, and in which the ion transport is negligible to obtain one-point calibration. Since one-point calibration in electrochemistry has a strict connotation that refers to fast calibration check, we change the terminology more appropriately to “base-line potential stability check”. We used this procedure to check the stability of the standard potential. Its drift that is provoked by the environment of our experiments during prolonged measurement could be a source of significant error. The check was to ensure that this does not happen. Afterwards, a direct determination of the main ion, which is in the linear Nernstian range, with the CV being as low as 0.4% for monovalent ions and meter resolution 0.1 mV is possible. We allowed for the error of 15%, so that the experiments were on the safe side. The ISE responses are Nernstian in the concentration ranges of analyte of interest in the matrices of the medium used. In the case of highest sodium to potassium ratio for KSH2, the interference of potassium adds 0.8 to 24.8 mM of sodium, which accounts for 3% increase in sodium. This results in a much smaller error than is allowed by us in biological research, where the total error permitted is +/−15% [24].

Recently, the micro-needle ion selective electrode was used for intradermal potassium detection [29]. Since the micro-needle electrode is in direct contact with living cells, the cytotoxicity test was performed. After 24 h of fibroblasts contact with the electrode, the considerable cytotoxicity that was observed was likely due to valinomycin leak from the electrode material. On the contrary to [29], electrodes are never in direct contact with cells, the medium bathing cells lacks proteins or lipids that are able to extract valinomycin from the ISE membrane, and the medium is replaced every 10 min by a fresh one in our experimental setup. We also systematically tested the viability of cells and the transepithelial resistance after series of measurements and found neither the change in transepithelial resistance nor a toxic effect on the epithelial cell layer.

## 3. Results

We studied the ion transport that was caused by the introduction of a sodium or chloride gradient across the epithelial cell layer. Four different pairs of media were used to form the concentration gradients: KHS1 (high sodium and high chloride) on the basolateral side of the epithelial cell layer and KHS2 (low sodium) on the apical side—Figure 4A; KHS1 medium on the apical side and KHS2 on the basolateral side—Figure 4B; KHS1 on the basolateral side and KHS3 (low chloride) on the apical side—Figure 5A, and KHS1 on the apical side and KHS3 on the basolateral side—Figure 5B. The results for sodium, chloride, and potassium transport are expressed as the change of ion concentration on a particular side of the epithelial layer 10 min after stopping the media flow. The pH values are also presented. The media acidifies from the original value of pH = 7.40, despite the medium composition and the side of the epithelial cell layer. The transepithelial potential difference was measured against the basolateral medium.

An unequally distributed ion species is transported down the electrochemical gradient, and the counter ion follows what is manifested by the transepithelial potential value that follows the gradient of ions, i.e., it is negative on the higher concentration of sodium ions and positive on the higher concentration of the chloride ions. The transepithelial potential value is less than 3 mV. Potassium transport plays a minor role in the observed transport. There is an obvious difference between the transported cations and anions.

## 4. Discussion

### 4.1. Medium pH Change Can Be Caused by Different Mechanisms

The pH value changed from the original 7.40 to 7.14 ± 0.02 in our experimental conditions in all media that were used and on both sides of the epithelial cell layer. There are three possible mechanisms that lead to media acidification: proton export, bicarbonate import, and carbon dioxide production. To change the pH value of KHS1 solution from 7.40 to 7.13, one needs to add 2.3 mM of HCl, or add 1.95 mM of CO_2_ (8.1% partial pressure of CO_2_ in equilibrium with the medium), or subtract 6.9 mM of NaHCO_3_. While NHE exchanger and/or bicarbonate secretion can cause medium acidification, there is no open route for bicarbonate import from the KHS1 medium on the apical side. We also used amiloride, the potent NHE exchanger blocker and DIDS potent blocker of AE exchanger, and found that they have no effect on acidification of the media. Thus, CO_2_ that is produced by cells caused acidification.

### 4.2. Medium Composition Could Affect the Direction of Transport of Biological Molecules

In classic electrochemistry, one tests a sample response by changing the medium composition. When studying living cells, the change of medium composition might completely alter the studied sample. The very direction of ion flux across the epithelium depends on the media composition. The direction of ion flux is given by the difference between the electrochemical potentials on both sides of the cell membrane for each ion. For the *X* ion, this is given by the equation:
$$\Delta \bar{\mu_{x}}=-RTln\left(\frac{\left[X_{c}\right]}{\left[X_{m}\right]}\right)+zF\left(\varphi_{m}-\varphi_{c}\right)$$ where subscripts *c* and *m* denote the cytoplasm and medium (either basolateral *b* or apical *a*), and *R*, *T*, *F*, *z* denote the gas constant, temperature, Faraday constant, and ion charge, respectively. If one adopts the following parameter values: *φ_m_*≡ 0 and *φ_c_*= −60 mV and for anion concentrations [Cl]_c_ = 40 mM [30], [HCO_3_]_c_ = 15 mM and [K]_c_ = 120 mM [31], [Na]_c_ = 30 mM [32]; KHS1 values [Cl]_m_ = 129.1 mM, [HCO_3_]_m_ = 24.8 mM, [Na]_m_ = 141.8 mM, [K]_m_ = 5.9 mM; KHS2 values differ from KHS1 in sodium concentration [Na]_m_ = 24.8 mM, and KHS3 with chloride concentration [Cl]_m_ = 10 mM, one can calculate the electrochemical potential differences for particular ions, e.g., in the case of the KHS1 solution $\Delta \bar{\mu_{Cl}}=-2.9\;\mathrm{kJ}/\mathrm{mol}$ and $\Delta \bar{\mu_{HCO3}}=-4.6\;\mathrm{kJ}/\mathrm{mol}$, which favours the exchange of bicarbonate out for chloride into the cytoplasm, while for KHS3 solution $\Delta \bar{\mu_{Cl}}=-9.3\;\mathrm{kJ}/\mathrm{mol}$ and $\Delta \bar{\mu_{HCO3}}=-4.6\;\mathrm{kJ}/\mathrm{mol}$ chloride will be exported from the cytoplasm in exchange for bicarbonate. Figure 6 shows the effect of the medium composition changes on the energetics and direction of action of different exchangers and transporters is shown. It is likely that the change of the direction of transported ions on the side of epithelial cells facing the low concentration medium might also lead to the entire cytoplasm composition change. The opening and closing of transporting proteins are under the control of the cell’s physiology. In our experiments, we do not measure the membrane potential of cells *φ_c_* and its changes. The membrane potential can be affected by ion transport as well as affecting ion transport. The chloride and bicarbonate ions flow-through anion channels can exemplify the effect of membrane potential change. For *φ_c_*= −60 mV, both bicarbonate and chloride ions flow out from the cell, for *φ_c_*= −30 mV, only bicarbonate flow out and there is no net chloride transport, for *φ_c_*= −20 mV bicarbonate flow out and chloride flow in, while for *φ_c_*= −10 mV both bicarbonate and chloride flow into the cell [33].

### 4.3. Isosmotic NaCl Flow Through Epithelial Cell Layer in Both Directions

The data presented in Figure 4 and Figure 5 show that the measured transport of cations is not equal to the anions, i.e., the transport is apparently not electroneutral, which is false. The simple explanation that the export or import of bicarbonate anion provides the balance between cations and anions does not hold, since bicarbonate transport should also affect pH. Thus, another explanation is needed.

Assuming that: (1) in the presence of an isosmotic gradient of sodium or chloride, there is an isosmotic transport of considerable volume of isosmotic 145 mM NaCl solution; (2) that the missing volume on one side is replaced by original medium; and, (3) that on the other side the newly appearing medium pushes out the original medium from the measuring chamber, one can calculate the final expected sodium and chloride concentrations and then compare them with the measured values. Figure 7 presents the experimental and theoretical results. The fit is surprisingly good. Thus, one can safely assume that, in our experiment, there is isosmotic volume flow from one side of the cell layer to the other in the response of sodium or chloride concentration gradient across the epithelial layer.

## 5. Conclusions

The measurement of the transport of many ions across the epithelial cell layer provides insight into the mechanism of epithelial physiology in health and disease. The biological cell system is very complicated from the electrochemical point of view. The change of ion composition of the medium can reverse the direction of ion transport. Measurement of the transmembrane ion fluxes is also a challenge to electroanalytical chemistry. In this report, we described a system for the concurrent measurement of sodium, potassium, chlorides, and pH in the vicinity of both sides of the epithelial cell layer. To get a full insight into the physiological processes, one needs to add a bicarbonate-sensitive ISE electrode to our apparatus. Only very recently, progress towards such an electrode has been reported by one of the authors [34]. One should also measure the volume transport across the epithelial layer, and we are now testing such a device. It is also necessary to measure the membrane potential of cells in the epithelial layer by means of this method, which will not disturb the physiology of the epithelium. Using our ISE-based apparatus, we found that, in spite of osmotic equilibrium, the ions are actively transported through an epithelial cell layer in response to a sodium or chloride gradient, which was often doubted. The transport of ions occurs in both directions in the same line of cells, in contrast to the idea that transport in each direction requires different types of cells. The obtained results fit to the hypothesis that, in sodium or chloride gradient, there is volume transport of 145 mM of NaCl.

## Figures and Tables

**Figure 1 sensors-19-01881-f001:**
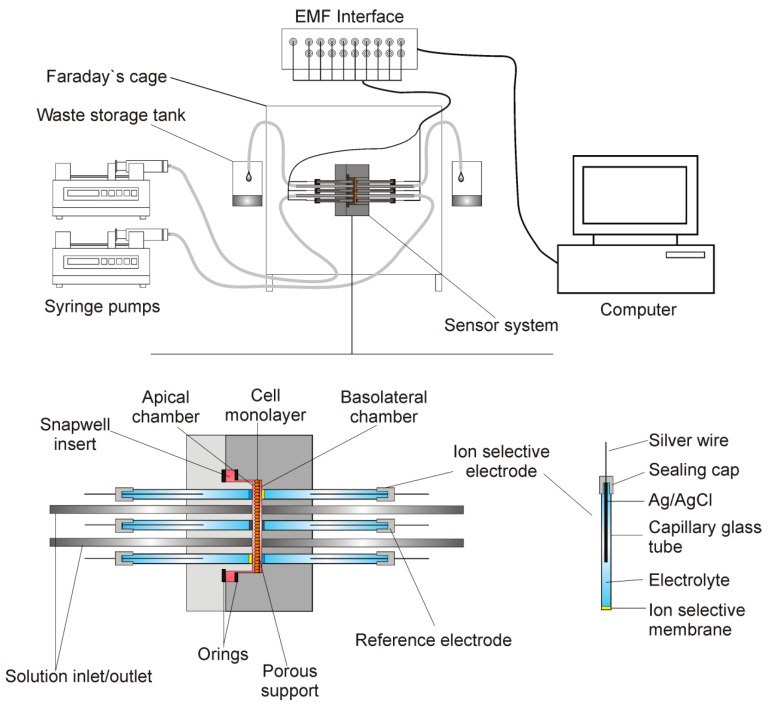
The measuring system. Dismountable measuring platform houses electrodes, inlet and outlet tubes and cells grown on the porous support. Exchangeable ion-selective electrodes (ISE) and reference electrodes are connected to the EMF (electromotive force) interface. The media are exchanged by two syringe pumps.

**Figure 2 sensors-19-01881-f002:**
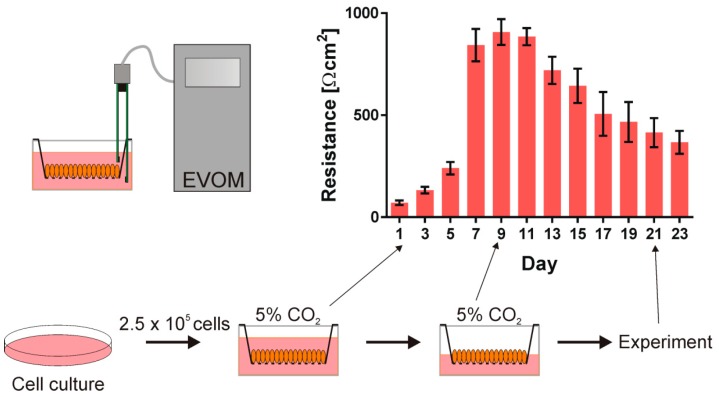
The human bronchial epithelial cell line (16HBE14σ) was grown in a plastic Petri dish in the humidified incubator at 37 °C in the presence of 5% CO_2_. Subsequently, the cells were seeded onto Corning Costar Snapwell inserts. First the cells were grown submerged in culture medium and their transepithelial resistance (TER) was monitored. When the cell layer reached the resistance of about 1000 Ω·cm^2^, the air contact at the apical side of the monolayer was introduced and after 10–16 days the cells were used in the experiment.

**Figure 3 sensors-19-01881-f003:**
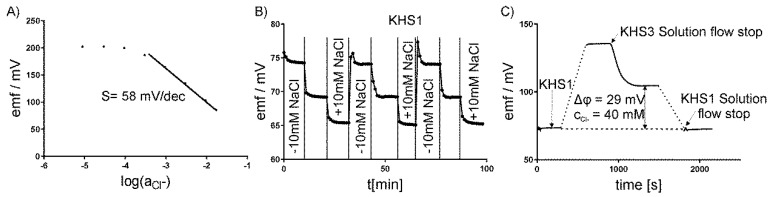
Calibration of ISE–chloride-sensitive electrode. (**A**) The calibration curve. (**B**) The potential reproducibility using Krebs–Henseleit solution (KHS1) lacking 10 mM of NaCl, KHS1, and KHS1 supplemented with 10 mM of NaCl. (**C**) One-point calibration procedure of a base-line potential stability check in the KHS1 medium.

**Figure 4 sensors-19-01881-f004:**
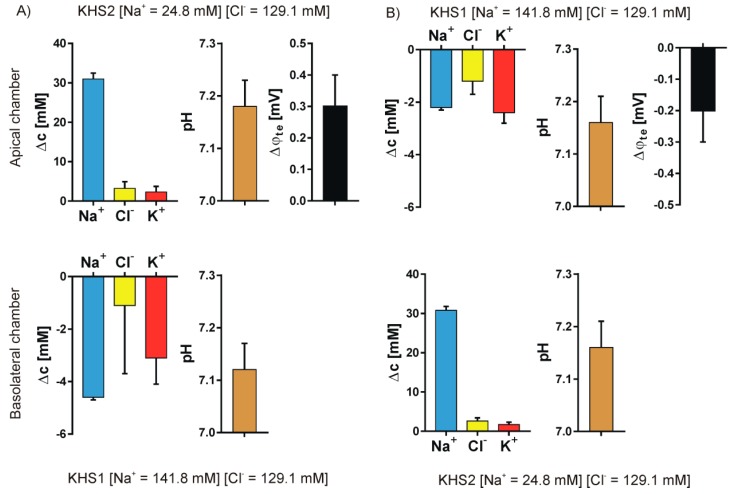
Sodium gradient. The results for sodium, chloride, and potassium transport are expressed as the change of ion concentration on a particular side of the epithelial layer 10 min after stopping the media flow, the pH value and the transepithelial potential value measured against the basolateral side. (**A**) KHS1 on the basolateral side of the epithelial cell layer and KHS2 on the apical side. (**B**) KHS1 medium on the apical side and KHS2 on the basolateral side.

**Figure 5 sensors-19-01881-f005:**
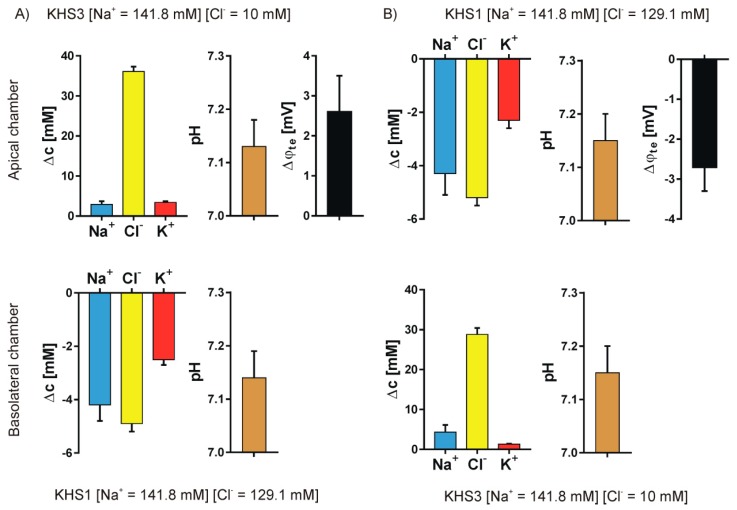
Chloride gradient. The results for sodium, chloride, and potassium transport are expressed as the change of ion concentration on a particular side of the epithelial layer 10 min after stopping the media flow, the pH value and the transepithelial potential value measured against the basolateral side. (**A**) KHS1 on the basolateral side of the epithelial cell layer and KHS3 on the apical side. (**B**) KHS1 medium on the apical side and KHS3 on the basolateral side.

**Figure 6 sensors-19-01881-f006:**
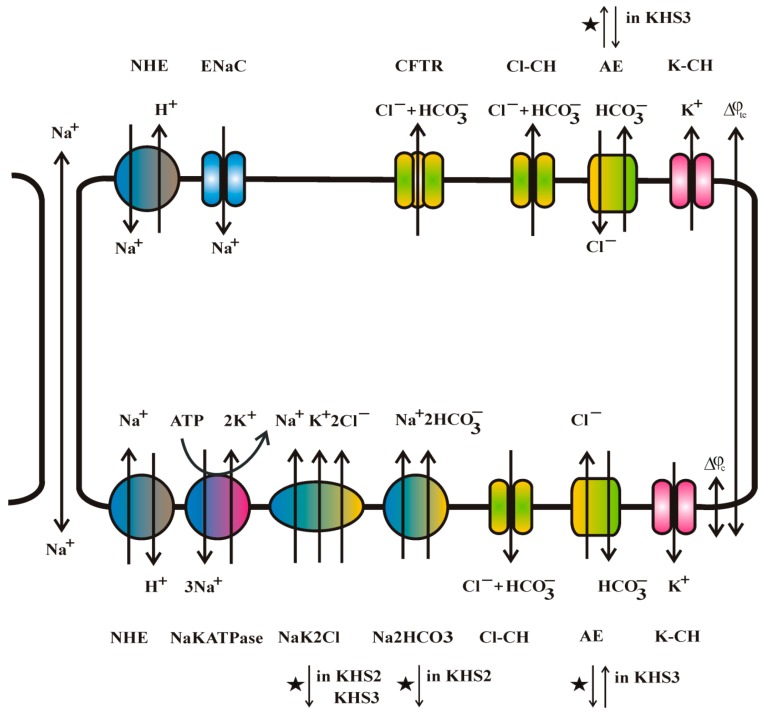
Major ion transporters of bronchial epithelium and the direction of ion flow. Sodium can be transported via a paracellular route. The sodium-transporting Epithelial Sodium Channel (ENaC) channel and anion-selective Cystic Fibrosis Transport Regulator (CFTR) channel are present on the apical side of the cell membrane. AE anion and NHE sodium for proton exchangers are present on both sides of the cell layer. NaKATPase, NaK2Cl and Na2HCO3 transporters are present on the basolateral side of the cell membrane. The other anion channels Cl-CH and potassium channels K-CH are present on both sides of the cell membrane. The action of ion-transporting molecules is controlled by many different factors, i.e., pH, cell volume, calcium concentration, cyclic nucleotides, and membrane potential. Its interrelation is not yet fully understood. Star denotes the change of transport direction in response to our experimental media KHS2 or KHS3.

**Figure 7 sensors-19-01881-f007:**
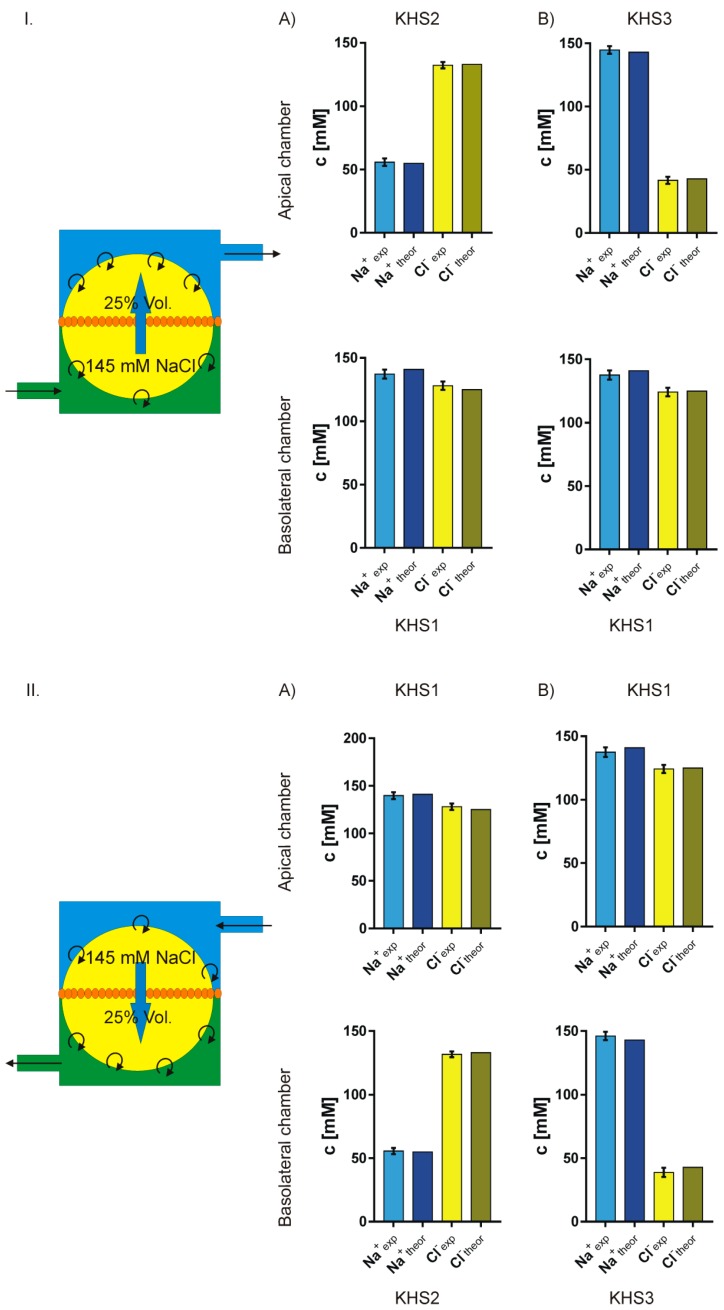
Comparison of the measured and theoretically predicted concentrations of sodium and chloride on both sides of the cell layer. The theoretical model illustrated on the left of the figures assumes that there is an isosmotic transport of 25% of chamber volume of 145 mM NaCl solution from one side of the cell membrane to the other. **I**. (**A**,**B**) lower concentration of sodium or chloride respectively on the apical side. **II**. (**A**,**B**) lower concentration of sodium or chloride respectively on the basolateral side.

**Table 1 sensors-19-01881-t001:** Composition of media used. All of the media contained [K^+^] = 5.9, [Mg^2+^] = 1.2, [Ca^2+^] = 2.5, [HCO_3_
^−^] = 24.8, [HPO_4_
^2−^] = 1.2, glucose = 11.1 and different concentrations of [Na^+^], choline [Chol^+^], [Cl^−^] and gluconate [Gluc^−^]. All solutions had the osmolality of 289 mOsm/kg. The pH = 7.4 value was obtained by saturating the solutions with carbogen (5% CO_2_, 95% O_2_). All concentrations in mM.

	KHS1	KHS2	KHS3
[Na^+^]	141.8	24.8	141.8
[Chol^+^]	0	117.0	0
[Cl^−^]	129.1	129.1	10
[Gluc^−^]	0	0	119.1

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
