# Peer review of "New ISE-Based Apparatus for Na+, K+, Cl−, pH and Transepithelial Potential Difference Real-Time Simultaneous Measurements of Ion Transport across Epithelial Cells Monolayer–Advantages and Pitfalls"

_sensors, 2019, doi:10.3390/s19081881_

Round 1
Reviewer 1 Report
The paper by Miroslaw et al. presents a device for the monitoring of ion transport across epithelial cells by using potentiometric sensors. While I am not able to judge the data interpretation because this is out of my field of expertise, I am going to provide my opinion about the potentiometry measurements per se.
I have two main concerning points, one regarding the one-point calibration method and the other regarding the sensor selectivity.
First, I do not fully understand why the membranes cannot be washed. Could you please explain this fact more in detail. Then, the authors should demonstrate that the one-point caliobration is suitable for a reliable calculation of the ion concentration by control experiments (maybe just the electrodes outside the device) and also give some stimation of possibles errors coming from this calibration process.
Regarding the selectivity:
1. Why you are using Na ionophore III instead of X? The one that you are using suffers from stronger K potassium than ionophore X.
2. Selectivity coefficitiens should be calculated for all the sensors and considered in the calculations of ion concentrations by using the Nikolskii-Einsenman equation.
3. While the interference of choline and gluconate is mentioned for the Cl sensors (as a change in the slope) this is not considered in the later calculations. Then, what about the other anions? Bicarbonate for instance? Please, calculate the selectivity coefficients and used them for your final calculations.
4. Also, I am not sure about the use of valinomycin to sense K, as it was already demonstrated that it has a toxic effect for human cells. Please, have a look to the recent paper by Parilla et al. (Anal. Chem. 2019, 91, 2, 1578-1586), include it in your references and comment on this possibility.
Other things that should be improved in the manuscript:
1. English correction.
2. Please, rewrite the introduction. There is now flow and it is plenty of unconnected sentences and stetements.
Author Response
Reviewer 1.
First, I do not fully understand why the membranes cannot be washed. Could you please explain this fact more in detail.
Epithelial cells respond to medium composition change by transporting ions and water across the cell layer. Thus in the presence of cells no testing of electrodes is possible because composition of the testing medium will be changed by cells.
Then, the authors should demonstrate that the one-point caliobration is suitable for a reliable calculation of the ion concentration by control experiments (maybe just the electrodes outside the device) and also give some stimation of possibles errors coming from this calibration process.
We used KHS1 solution similar to the one cells were grown in and in which the ion transport is negligible to get one point calibration. Since one point calibration in electrochemistry has a strict connotation that refers to fast calibration check we change the terminology to more adequate, namely “base-line potential stability check”
We use the classical membranes. They were described in hundreds of papers. Their stabile sensitivity combined with sufficient (for our experiments) selectivity and detection limits justifies the application of the Nernstian approach and one-point calibration. Single point calibration adopted by us is used to check the stability of the standard potential. Its drift provoked by the environment of our experiments during prolong measurement could be a source of significant error. The check was to ensure that this does not happen. Then, a direct determination of the main ion, in the linear Nernstian range, with the CV as low as 0.4% for monovalent ions and meter resolution 0.1 mV is possible. We allowed for the error of 15% so the experiments were of the safe side.
Regarding the selectivity:
1. Why you are using Na ionophore III instead of X? The one that you are using suffers from stronger K potassium than ionophore X.
Thanks for this comment. We used ETH 2120, ETH 4120 and Ionophore X. Indeed, the selectivity to potassium of the ionophore is half an order better (in -log scale) than for remaining ligands. However, we selected ETH 2120 since the membranes with this ionophore provides sufficient selectivity in the course of our experiments (-log KNa,K = 1.5) and excellent stability over several months. Additionally, our approach is supported by log-term industrial experience (AL) with this ionophore in the environment of commercial analyzers (see: Lewenstam, A. Routines and Challenges in Clinical Application of Electrochemical Ion- Sensors. Electroanalysis, 2014, 26, 1171–1181) The ISE membrane with Ionophore X is not stable enough. It loses its properties after few hours.
2. Selectivity coefficitiens should be calculated for all the sensors and considered in the calculations of ion concentrations by using the Nikolskii-Einsenman equation.
In the case of cations, being aware of this equation, we have checked that the ISE responses are Nernstian in the concentration ranges of analyte of interest in the matrices of the medium used measuring the responses characteristic for the main ions we were sure that the terms CNa>>KNa,K CK In the case of highest sodium to potassium ratio for KSH2 the interference of potassium adds 0.8 to 24.8 of sodium which accounts for 3% increase in sodium. This results in much smaller error than allowed by us in biological research where the total error permitted is +/-15 % (see: Zajac, M.; Lewenstam, A.; Dolowy, K. Multi-electrode system for measurement of transmembrane ion-fluxes through living epithelial cells. Bioelectrochemistry 2017, 117, 65-73).
3. While the interference of choline and gluconate is mentioned for the Cl sensors (as a change in the slope) this is not considered in the later calculations. Then, what about the other anions? Bicarbonate for instance? Please, calculate the selectivity coefficients and used them for your final calculations.
Thank you for the question. We clarified the testing procedure which now reads:
The potential reproducibility and empirical calibration curve was determined in our system lacking a cell layer by repeating the medium exchange for different concentrations of measured ion. Exemplary testing of the chloride-sensitive electrode in three KHS1 media lacking 10 mM of NaCl, the original KHS1 and KHS1 supplemented with 10 mM of NaCl is presented in Figure 3B. The same procedure was repeated for KHS2 and KHS3 media supplemented or lacking NaCl. All electrodes were tested by the same procedure. The media compositions for the sodium-sensitive electrode were the same as for the chloride-sensitive electrode, for the potassium-sensitive electrode the media were supplemented with 2mM of KCl (or lacking 2mM of KCl), and for the pH-sensitive electrode pH=7.2, 7.4 and 7.6 were used.
The use of choline and gluconate ions have adverse effects on sodium-sensitive or chloride-sensitive ISEs. The slope for the chloride ISE in the presence of choline was decreased to s = –40±2 mV/dec. in KHS2 medium and for the sodium ISE in the presence of gluconate to s = +40±2 mV/dec. in KHS3 solution. The apparent sensitivity observed in KHS2 and KHS was encountered in calculations of concentration changes in these solutions.
ISEs are often used for automated determination of ion concentrations in whole blood or plasma. In such measurements, the electrode is usually washed and calibrated before each measurement within a few seconds [27,28]. This could not be done in our experiment, since the washing solution is in direct contact with the cell layer, which influences its ion transport properties and the experiments last several minutes without the possibility of the medium change. Therefore, the base-line potential stability check, reassembling the procedure known as one-point calibration, was employed. First, each ISE potential in the KHS1 solution was measured. Then, after changing KHS1 to the other medium the potential measured and the electrode slope were used to calculate the concentration (or concentration change) of a particular ion. The values of potential after 10 minutes were taken for the calculations. The base-line check was closed repeating the measurement in KHS1, for graphical illustration see Figure 3C. After the series of experiments, the cell layer transepithelial resistance and viability was tested and no significant changes were observed.
In the case of chloride ISE and the influence of choline and gluconate the apparent slopes were assessed and used to calculate concentration changes.
Bicarbonate and gluconate interference is process-wise similar to that of potassium ion in the case of sodium ISE. With the selectivity coefficients for these anions in the order of 0.1 and the constant ionic background in KHS media we found that apparent sensitivity approach is sufficient for obtaining biologically meaningful results.
4. Also, I am not sure about the use of valinomycin to sense K, as it was already demonstrated that it has a toxic effect for human cells. Please, have a look to the recent paper by Parilla et al. (Anal. Chem. 2019, 91, 2, 1578-1586), include it in your references and comment on this possibility.
Valinomycin is a powerful antibiotic studied extensively since 1960th. It transports selectively potassium ions through biological membranes. It is extremely toxic. The intake of a few miligrams of valinomycin will be fatal. However, incorporation of valinomycin into the hydrophobic ISE membrane restricts its ability to dissolve in water and to affect cells. The following sentence was added to the text:
After series of experiments cell layer transepithelial resistance and viability was tested and no significant changes were observed.
The other problem is Parrilla (!) et al. 2019 excellent paper. They used miniaturized valinomycin based solid state microelectrode for intradermal potassium detection. Paper is excellent but it was preceded by hundreds of papers on the same subject published in last 40 years since potassium concentration detection must be constantly monitored during major surgery procedure. The subject is worth a review with many citations and not mentioning only the last paper even as good as Parrilla’s one.
Other things that should be improved in the manuscript:
1. English correction.
It is improved
2. Please, rewrite the introduction. There is now flow and it is plenty of unconnected sentences and stetements.
Introduction is rewritten.
Reviewer 2 Report
Author should compare this work with similar work such as follow:
1. Analyst. 2013 Aug 7;138(15):4292-7. doi: 10.1039/c3an00843f. Epub 2013 Jun 18.
Electrochemical detection of chloride levels in sweat using silver nanoparticles: a basis for the preliminary screening for cystic fibrosis
2. Biosens Bioelectron. 2009 Feb 15;24(6):1788-91. doi: 10.1016/j.bios.2008.07.051. Epub 2008 Aug 3.
Early determination of cystic fibrosis by electrochemical chloride quantification in sweat.
Author Response
Author should compare this work with similar work such as follow:
1. Analyst. 2013 Aug 7;138(15):4292-7. doi: 10.1039/c3an00843f. Epub 2013 Jun 18.
Electrochemical detection of chloride levels in sweat using silver nanoparticles: a basis for the preliminary screening for cystic fibrosis
2. Biosens Bioelectron. 2009 Feb 15;24(6):1788-91. doi: 10.1016/j.bios.2008.07.051. Epub 2008 Aug 3.
Early determination of cystic fibrosis by electrochemical chloride quantification in sweat.
Since the chloride elevation in cystic fibrosis was found in 1940th numerous papers on chloride detection in sweat were published. Maybe the subject is worth review paper with hundreds of citations. ISE electrodes are used in chloride in sweat detection in newborns since 1970th. Toh et al, 2013 in Analyst or Gonzalo-Ruiz et al., 2009 in Biosensors & Bioelectronics are interesting papers but only two of the vast body of publication on the matter and it would be inappropriate to mention these two papers and neglect 400 others (Web of Science). Moreover, cells we are using are not producing sweat but are responsible for maintaining thin fluid layer on a surface on the lungs.
Reviewer 3 Report
The submitted manuscript represents a very interesting piece of research and I believe it can be accepted for publication in Sensors after a minor revision addressing the following issues:
- both abstract and conclusion have to highlight the particular findings of the study; at the moment both of these sections only contain the statements on the general difficulty of the problem which is of course appreciated, but doed not give a hint to the reader on what was done in the study;
- please also highligh the sensor-related aspects of the study in these sections;
- please improve the quality of the Fig.3 - at the moment it is too small to see the details;
- page 6 line 173 - I beleive PVC is a more common abbreviation;
- page 6 line 192-193 - please correct the double mentioning of chloride;
- please make sure that all abbreviations Like NHE, DIDS, etc are decoded through teh text of teh manuscript;
- please perform thorough check of the text as there are many small inconsistencies like upper/lower indexes, English grammar, etc.
Author Response
The submitted manuscript represents a very interesting piece of research and I believe it can be accepted for publication in Sensors after a minor revision addressing the following issues:
- both abstract and conclusion have to highlight the particular findings of the study; at the moment both of these sections only contain the statements on the general difficulty of the problem which is of course appreciated, but doed not give a hint to the reader on what was done in the study;
Thank you for your comment. You are right. We rearranged text in Abstract and Conclusion. We added to Abstract the following sentences:
Using flat miniaturized ISE electrodes allows reducing the medium volume adjacent to cells to approximately 20 μl and detecting changes in ion concentrations caused by transport through the cell layer.
and
We found that, in the isosmotic transepithelial concentration gradient of sodium or chloride ions, there is an electroneutral transport of sodium chloride in both directions of the cell monolayer. The ions and water are transported as isosmotic solution of 145mM of NaCl.
And to the Conclusions
Using our ISE based apparatus we found that in spite of osmotic equilibrium ions are actively transported through an epithelial cell layer in response to sodium or chloride gradient what was often doubted. The transport of ions occurs in both directions in the same line of cells on the contrary to the idea that transport in each direction requires different type of cells. The results obtained fit to the hypothesis that in sodium or chloride gradient there is volume transport of 145mM of NaCl.
- please also highligh the sensor-related aspects of the study in these sections;
Done
- please improve the quality of the Fig.3 - at the moment it is too small to see the details;
Done
- page 6 line 173 - I beleive PVC is a more common abbreviation;
Done
- page 6 line 192-193 - please correct the double mentioning of chloride;
Done
- please make sure that all abbreviations Like NHE, DIDS, etc are decoded through teh text of teh manuscript;
Done
- please perform thorough check of the text as there are many small inconsistencies like upper/lower indexes, English grammar, etc.
Done
Round 2
Reviewer 1 Report
1) This reviewer really appreciate the improvement of the paper.
2) But, it is a pitty that the authors opted to answer the raised concerns without introducing the corresponding comments in the manuscript. In this sense, could you please introduce the following comments into the tex in an appropriate way?
“We used KHS1 solution similar to the one cells were grown in and in which the ion transport is negligible to get one point calibration. Since one point calibration in electrochemistry has a strict connotation that refers to fast calibration check we change the terminology to more adequate, namely “base-line potential stability check”
”Single point calibration adopted by us is used to check the stability of the standard potential. Its drift provoked by the environment of our experiments during prolong measurement could be a source of significant error. The check was to ensure that this does not happen. Then, a direct determination of the main ion, in the linear Nernstian range, with the CV as low as 0.4% for monovalent ions and meter resolution 0.1 mV is possible. We allowed for the error of 15% so the experiments were of the safe side.”
”We used ETH 2120, ETH 4120 and Ionophore X. Indeed, the selectivity to potassium of the ionophore is half an order better (in -log scale) than for remaining ligands. However, we selected ETH 2120 since the membranes with this ionophore provides sufficient selectivity in the course of our experiments (-log KNa,K = 1.5) and excellent stability over several months. Additionally, our approach is supported by log-term industrial experience (AL) with this ionophore in the environment of commercial analyzers (see: Lewenstam, A. Routines and Challenges in Clinical Application of Electrochemical Ion- Sensors. Electroanalysis, 2014, 26, 1171–1181) The ISE membrane with Ionophore X is not stable enough. It loses its properties after few hours.”
“In the case of cations, being aware of this equation, we have checked that the ISE responses are Nernstian in the concentration ranges of analyte of interest in the matrices of the medium used measuring the responses characteristic for the main ions we were sure that the terms CNa>>KNa,K CK In the case of highest sodium to potassium ratio for KSH2 the interference of potassium adds 0.8 to 24.8 of sodium which accounts for 3% increase in sodium. This results in much smaller error than allowed by us in biological research where the total error permitted is +/-15 % (see: Zajac, M.; Lewenstam, A.; Dolowy, K. Multi-electrode system for measurement of transmembrane ion-fluxes through living epithelial cells. Bioelectrochemistry 2017, 117, 65-73).”
3) Despite your explanation concerning the valinomycin, you did not state the possible toxicity with the cells that you used in your paper. I perfectly know the extended use of vanilomycin as ionophore, indeed this is not my concern, but I am refering to its toxicity effect when it is contact with the cells. In this sense, please introduce a comment about this issue refering to the paper by Parrilla et al. and maybe other papers averting from this issue. I understand that the membrane use by Parilla et al. is thinner that that used in your studies, but this is the only paper showing that the valinomycin leaching from a plasticized polymeric membrane causes toxic effects in cells while running potentiometric measurements. In my opinion, this has to be comment on your text ,as you are running analogous measurements with cells. Obviously, for a better evaluation of this effect, you should accomplish toxicity experiments under your specific experimental conditions. I am not asking to that, but just to mention on this possibility on your paper.
Author Response
1) This reviewer really appreciate the improvement of the paper.
Thank you. Some additional improvements has been done
2) But, it is a pitty that the authors opted to answer the raised concerns without introducing the corresponding comments in the manuscript. In this sense, could you please introduce the following comments into the tex in an appropriate way?
Done.
3) Despite your explanation concerning the valinomycin, you did not state the possible toxicity with the cells that you used in your paper. I perfectly know the extended use of vanilomycin as ionophore, indeed this is not my concern, but I am refering to its toxicity effect when it is contact with the cells. In this sense, please introduce a comment about this issue refering to the paper by Parrilla et al. and maybe other papers averting from this issue. I understand that the membrane use by Parilla et al. is thinner that that used in your studies, but this is the only paper showing that the valinomycin leaching from a plasticized polymeric membrane causes toxic effects in cells while running potentiometric measurements. In my opinion, this has to be comment on your text ,as you are running analogous measurements with cells. Obviously, for a better evaluation of this effect, you should accomplish toxicity experiments under your specific experimental conditions. I am not asking to that, but just to mention on this possibility on your paper.
The following discussion is added to our text citing the paper of Parrilla et al.:
Recently the micro-needle ion selective electrode was used for intradermal potassium detection [29]. Since microneedle electrode is in direct contact with living cells cytotoxicity test was performed. After 24 hours of fibroblasts contact with electrode the considerable cytotoxicity was observed likely due to valinomycin leak from the electrode material. On the contrary to [29] in our experimental setup electrodes are never in direct contact with cells, the medium bathing cells lacks proteins or lipids able to extract valinomycin from the ISE membrane and medium is replaced every 10 minutes by fresh one. We also systematically tested viability of cells and the transepithelial resistance after series of measurements and found neither the change in transepithelial resistance nor toxic effect on epithelial cell layer.
Round 3
Reviewer 1 Report
Thanks for the large improvement of the manuscript. I recommend its acceptance in the present form. Good luck for your further investigations.